# “Real-Time Neuromonitoring” Increases the Safety and Non-Invasiveness and Shortens the Duration of Idiopathic Scoliosis Surgery

**DOI:** 10.3390/jcm13051497

**Published:** 2024-03-05

**Authors:** Przemysław Daroszewski, Juliusz Huber, Katarzyna Kaczmarek, Piotr Janusz, Paweł Główka, Marek Tomaszewski, Tomasz Kotwicki

**Affiliations:** 1Department of Organization and Management in Health Care, Poznań University of Medical Sciences, 28 Czerwca 1956 r. Street, No. 135/147, 61-545 Poznań, Poland; dyrektor@orsk.ump.edu.pl; 2Department Pathophysiology of Locomotor Organs, Poznań University of Medical Sciences, 28 Czerwca 1956 r. Street, No. 135/147, 61-545 Poznań, Poland; katarzynakaczmarek@ump.edu.pl; 3Department of Spine Disorders and Pediatric Orthopaedics, Poznań University of Medical Sciences, 28 Czerwca 1956 r. Street, No. 135/147, 61-545 Poznań, Poland; pjanusz@ump.edu.pl (P.J.); pawel.glowka@ump.edu.pl (P.G.); mtooma@wp.pl (M.T.); kotwicki@ump.edu.pl (T.K.)

**Keywords:** idiopathic scoliosis, intraoperative neurophysiological recordings, real-time neuromonitoring, transcranially evoked motor potentials, muscle versus nerve recording

## Abstract

**Introduction:** A practical solution to the incidental unreliability of intraoperative neuromonitoring (IONM) may be the simultaneous neurophysiological recording and control of the surgical field through a camera (the concept of “Real-time” IONM). During “Real-time” IONM, the surgeon is immediately warned about the possibility of damage to the neural structures during, but not after, standard idiopathic scoliosis (IS) corrective surgery procedures (the concept of “Surgeon–neurophysiologist” interactive, verbal IONM). This study aimed to compare the advantages, utilities, reliabilities, and time consumption of the two IONM scenarios. **Methods:** Studies were performed in two similar groups of patients undergoing surgery primarily due to Lenke 2 idiopathic scoliosis (N = 120), when both IONM approaches were applied. Neurophysiological evaluations of the spinal transmission were performed pre- (T0), intra- (before (T1) and after (T2) surgery), and postoperatively (T3), as well as once in healthy volunteers (control, N = 60). Non-invasive and innovative recordings of the motor evoked potentials (MEPs) bilaterally from the peroneal (PER) nerve and tibialis anterior (TA) muscle were performed with surface electrodes as a result of transcranial magnetic stimulation (TMS) or electrical stimulation (TES) at T0–T3. **Results:** In both groups, the MEP amplitudes and latencies recorded from the PER nerve were approximately 67% lower and 3.1 ms shorter than those recorded from the TA muscle. The MEP recording parameters differed similarly at T0–T3 compared to the control group. In all patients, the MEP parameters induced by TMS (T0) and TES (T1) did not differ. The MEP amplitude parameters recorded from the TA and PER at T1 and T2 indicated a bilateral improvement in the neural spinal conduction due to the surgical intervention. The TMS-induced MEP amplitude at T3 further increased bilaterally. In both IONM groups, an average 51.8 BIS level of anesthesia did not affect the variability in the MEP amplitude, especially in the PER recordings when the applied TES strength was 98.2 mA. The number of fluctuations in the MEP parameters was closely related to the number of warnings from the neurophysiologist during the transpedicular screw implantation, corrective rod implantation, and distraction, derotation, and compression procedures, and it was higher in the “Surgeon–neurophysiologist” IONM group. The average duration of surgery was shorter by approximately one hour in the “Real-time” IONM group. The number of two-way communications between the surgeon and the neurophysiologist and vice versa in the “Real-time” IONM group decreased by approximately half. **Conclusions:** This study proves the superiority of using “Real-time” IONM over the standard “Surgeon–neurophysiologist” IONM procedure in increasing the safety and non-invasiveness, shortening the time, and lowering the costs of the surgical treatment of IS patients. The modifications of the MEP nerve-conduction-recording technology with surface electrodes from nerves enable precise and reliable information on the pediatric patient’s neurological condition at every stage of the applied surgical procedures, even under conditions of slight fluctuations in anesthesia.

## 1. Introduction

Currently, the prevalence of pathological lateral spine curvature and its rotation may have reached 5–6% in a worldwide population of adolescents, and predominately in girls [1,2]. Idiopathic scoliosis (IS) is the spinal deformity that is most commonly treated surgically [3]. The conservative treatment with kinesiotherapy and bracing in the vast majority of girls with IS usually fails, and it may only diminish or slow down the curvature progression; still, it is the first-choice treatment [4,5,6,7]. Unfortunately, non-treated scoliosis may lead to neuropathies in the lower-extremity nerves with consequences such as neurogenic muscle injuries and, finally, paralysis or advanced cardiopulmonary diseases [8,9]. Spinal deformation surgery is an art of great complexity, and new methods with different possible iatrogenic risks are constantly being introduced [10]. It is obvious that a greater magnitude of the preoperative deformity and surgical extent increases the risk of spinal cord injury identified via intraoperative neuromonitoring (IONM) alerts during the correction of deformities in patients with IS [11]. It has been found that in about 13% of patients undergoing spinal deformity correction, it was necessary for the neuromonitoring team to alert the surgeon about the side effects of the surgery [12]. Moreover, the incidence of possible severe postoperative neurologic deficits has been assessed at about 3.2% for scoliosis surgery [13]. The successful surgical implantation of the pedicle screws and correction with the implanted corrective rods are fraught with risks that may worsen the patient’s neurological health status [14]. These risks mainly include direct spinal cord trauma or the consequences of its stretching during deformity correction, as well as ischemia and cardiopulmonary abnormalities [12].

Confidence in the neurophysiological monitoring that supports the proper proceeding of IS surgical correction has increased since the number of iatrogenic side effects became incidental [15,16]. The generally accepted guideline according to the statement of the International Society of Intraoperative Neurophysiology [17] is that neuromonitoring procedures should be performed by an experienced clinical neurophysiologist and not by the personnel in the theater, who are not experienced in IONM result interpretation. Several procedures for the response to neuromonitoring-related changes have been created, and they can be changed depending on the IS surgery modifications [18]. Modifications of IONM have introduced the possibility of improving, shortening, and increasing the reliability of the cooperation between the neurophysiologist and surgeon to obtain the best results from the scoliosis treatment [19]. The standards of neuromonitoring are constant and consist of checking every step of the surgical procedure and reacting to improper incidents [20]. Although a consensus-based checklist to guide the surgeon’s reactions to IONM changes and best-practice guidelines for their recording have been formulated [21], and their agreement has been confirmed and validated [22], new concepts have since been developed [23,24,25]. It is obvious that the surgical team of spine surgeons, anesthesiologists, and neurophysiologists cooperate more efficiently if their experience, technical skills, and equipment are advanced [26]. The best results are achieved when the threats are predictable and when the effects of the surgeon’s activities are assessed in real time not only neurophysiologically but also visually. Levin et al. [27] underline that the communication between the anesthesiologists, neurophysiologists, surgeons, and nursing personnel is essential to the effective use of IONM. If we consider that the elimination of the discussed events and actions during intraoperative warnings may shorten the reaction time of both the neurophysiologist and the surgeon [23,28], a practical way to solve this problem may be the simultaneous neurophysiological recording and inspection of the surgical field through the camera. This would immediately warn the surgeon during, but not after, the introduction of a certain procedure to the spine, following pedicle screw implantation or corrective rod mounting. Considering that scoliosis surgery often takes from 4 to 6 h [29], every attempt to make it shorter and simultaneously safer is of significant interest, especially for the patient, whose health status may be significantly influenced by the anesthesia duration.

The motor evoked potential (MEP) transcranially induced with magnetic field stimulation (TMS) is a very precise tool that is widely used for the evaluation of the motor function of patients with degenerative myelopathy [30] or incomplete spinal cord injuries [31,32]. The diagnostic sensitivity of the MEP in the detection of the spinal cord structure insult has been evaluated at 98% [33]. MEPs induced with the trains of transcranially applied electrical stimuli (TES) are characterized by an almost 100% sensitivity and specificity to the detection of corticospinal tract injury in the anterior and lateral spinal cord funiculi, as well as its ischemia consequences during spinal deformity correction [34]. Due to the high-amplitude potential recordings of more than 2000 μV, they do not need averaging. If the stimulus strength to induce the MEP is at 100 mA and the anesthesia level is kept stable, they are a reliable, affordable, and practical neuromonitoring tool during scoliosis correction [8]. Spinal cord motor function monitoring with the MEP is generally considered based on recordings of the muscle responses following the intermittent stimulation of the motor cortex in real time [20], which seems to be partially true. Currently, the “Real-time-neuromonitoring” concept is usually understood as the recording of the evoked potentials after certain surgical procedures that are applied to IS patients, verifying their non-invasiveness to the spinal cord neural transmission. The “Real-time-neuromonitoring” concept proposed in this paper refers to the recording of the evoked potentials during surgical corrective procedures applied to IS patients to verify that there is no insult to the spinal cord structures responsible for proper neural transmission. Simultaneous neurophysiological recording allows for the immediate reaction of the surgeon after warnings from the neurophysiologist.

During intraoperative neuromonitoring including the scoliosis surgery, bilateral recordings from the lower extremity muscles are used for the evaluation of the entire efferent neural transmission to the effector following the TES of the motor cortex centers. This allows for the evaluation of the supraspinal and intraspinal efferent pathways’ transmission and the neural transmission within the spinal ventral roots and motor fibers in the peripheral nerve [35,36]. The method is sensitive in the detection of motor deficits and the spinal cord reaction to ischemia intraoperatively. However, the amplitude parameter of the MEP is likely to be sensitive to the deep anesthesia influence, and the conditions of the recordings preclude the use of the constant neuromuscular blockade. An interesting proposal that seems to resolve the above problems is recording from the lower-extremity nerves [37], when the amplitude parameters are approximately half as low as those recorded from the muscles [38] but can be stably recorded, regardless of the level of anesthesia [23].

The main aim of this paper is to present details of the “Real-time” IONM concept to verify the safety of the procedures performed during the surgical correction of idiopathic scoliosis. To confirm its effectiveness, we compared the results of MEP recordings from the lower-extremity muscles versus those from the nerves following TMS and TES, respectively. They were performed in two equal groups of neuromonitoring sessions in IS patients with similar advancements in the pathology: the “Interactive S-N group”, based on bilateral surgeon–neurophysiologist verbal reports, and the “Real-time neuromonitoring group”, based on the simultaneous MEP recording and direct visual inspection of the operation field via the camera, without verbal reports. Our previous pilot studies on using “Real-time monitoring” in 35 cases of patients [23] allowed for a preliminary evaluation of the utility of this method concerning the total surgical procedure duration, which was verified in this study on a larger population of IS patients. In this study, the null hypothesis was that there is no difference in the advantages, usability, reliability, and time consumption between the IONM approaches.

## 2. Materials and Methods

### 2.1. Participants and Study Design

The basic research methods used in this work were pre-, double intra- (before and after scoliotic curvature correction), and postoperative recordings of the motor potentials evoked as a result of their transcranial induction, either with single magnetic field pulses (TMS) or a series of electric pulses (TES), leading to the excitation of the efferent pathways from the brain motor centers. The MEP parameters were analyzed bilaterally in recordings from the tibialis anterior (TA) muscle or from above the surface of the peroneal (PER) nerve, lateral to the head of the fibula, in the place of their longitudinal, anatomical course. While the choice of recording from the TA muscle was dictated by the possibility of comparing the MEP parameters with the descriptions of other researchers monitoring the overall efferent conduction from the level of the upper motoneurone to the effector, recording from above the nerve surface is innovative, previously described in studies on small populations of patients with locomotor dysfunction as a consequence of disc–root conflicts [38] and scoliosis [8]. The aim of describing it in this paper is to provide a detailed presentation of an alternative intraoperative neuromonitoring method, which is possibly more resistant to the anesthesia agent’s influence than recording from the muscle. To prove the superiority of one of the IONM methods (“Interactive S-N” vs. “Real-time” neuromonitoring), we selected, in three stages, two almost similar populations of scoliotic patients treated surgically with the same method (Figure 1).

From our database of patients with idiopathic scoliosis treated surgically for the first time between 2018 and 2023 (N = 377), we preliminarily selected results from motor-evoked-potential (MEP) recordings of girls with idiopathic scoliosis pre- (T0), intra- (before (T1) and after (T2) surgical procedures), and postoperatively (T3), as well as the detailed data included in the full IONM protocols collected from 298 subjects (Stage I). All patients were evaluated (including the analysis of anterior–posterior and lateral X-rays) and treated at Wiktor Dega Orthopedic and Rehabilitation Hospital in Poznań, Poland, by the same team of four surgeons. Two experienced neurophysiologists and two neurologists evaluated their health statuses pre- and postoperatively and performed the neuromonitoring procedures.

Applying the criteria of similar demographic, anthropometric, and scoliosis characteristics (type and curvature angle; Table 1), similar extents of the surgical approach from T1 to L2 from the back, and the same Nova Spine (Amiens, France) surgical corrective instrumentation (including a similar number of implanted transpedicular screws: from 8 to 16, 10 on average), we selected the records of 231 IS patients (Figure 1, Stage II). Then, in Stage III, based on the two different IONM scenarios but including similar MEP-recording conditions (bilaterally over the tibialis anterior (TA) muscle and from the surface of the peroneal (PER) nerve at the knee), as well as similar stimulus strengths to evoke the motor potentials during the TES (from 80 to 130 mA; mean: 98.2 ± 7.8 SD) and similar levels of applied anesthesia (bispectral index monitor (BIS): between 40 and 60) [39,40,41], we randomly allocated 146 patients to two equal groups of 60 patients each: the “Interactive S-N” group and the “Real-time” group.

Before the surgeries, all the treated girls belonging to the two studied groups had applied with no exceptions the Cheneau brace, and, in about half of the patients, physiotherapy exercises were prescribed to slow down the scoliosis progression.

Exclusion criteria for transcranial stimulation to induce the MEP when applied pre- and postoperatively or intraoperatively during the neuromonitoring included episodes of epilepsy; past brain lesions; skull defects; increased intracranial pressure; symptoms of cardiac and vascular diseases; the intake of proconvulsant medications or anesthetics; implanted intracranial electrodes, vascular clips, or shunts; and cardiac pacemakers or other implanted biomedical devices [35,36]. We followed the rules of IONM according to the guidelines of MacDonald [35].

A control group of 60 healthy girls was examined once to establish reference values for the neurophysiological recordings. The control group demographics (gender, age, height, and weight) were adjusted to match those of the studied groups. Statistically significant differences in age, height, and weight between the study groups and healthy controls were not observed (Table 1). The parameters of the amplitudes and latencies of the MEP evoked following TMS or TES were compared at each observation period in the patients belonging to the two study groups and healthy volunteers.

The study was approved by the Bioethics Committee of the University of Medical Sciences (Poznań, Poland; decision number 942/2021). Ethical considerations were in agreement with the Declaration of Helsinki (including the studies on healthy people). Each subject or her parent/legal guardian provided written consent for the examinations and the medical data publication, which were kept confidential.

### 2.2. Anesthesia, Spine Surgery, Neurophysiological Recordings, and Neuromonitoring Principles

The surgeries on the scoliotic patients were performed under propofol/remifentanil anesthesia (induction dose of remifentanil: 0.5 µg/kg; propofol: 2 mg/kg, and later, remifentanil: 0.5–2.0 µg/kg/h; propofol: 2–4 mg/kg/h, in continuous infusion) with a one-time dose of neuromuscular blockade (0.5 mg/kg of rocuronium bromide) at the beginning of the procedure. The level of anesthesia was monitored continuously in the bispectral index monitor (BIS) (GE Healthcare, Helsinki, Finland). Its level was kept constant from 40 to 60 during all the applied surgery procedures and neuromonitoring MEP recordings [39]. Blood pressure (maintained between 80 and 100 mmHg), temperature, %SpO_2_, and CO_2_ partial pressure were continuously monitored and maintained within their physiological limits. Inhalational anesthetics were not routinely used [44]. All anesthetic procedures were performed by the same two experienced anesthesiologists.

The implantation of a Nova Spine corrective instrumentation system (Amiens, France; Figure 2(Cc) and Figure 3I,J) was applied via a posterior approach to patients in a prone position during the scoliotic spine surgery (Figure 2(Ca)). The spine area from the upper thoracic vertebrae to the lower lumbar vertebrae was prepped and draped. When the posterior midline skin incision was performed, the paraspinal muscles were then dissected subperiosteally. The spine was exposed bilaterally from the midline along the spinous processes, from the laminas to the tip of the transverse processes (Figure 2(Cb)). To control bleeding, cauterization of the paravertebral muscles was necessary (Figure 2(Cb)). For the final anatomical wound closure, the spinous processes with the supraspinous ligament were preserved. The removed pieces of bones from the processes and released spine joints were used as the autografts for the final fusion following the bilateral application of 8–16 transpedicular screws (10 on average in each of the patients) with the free-hand technique for mounting the two corrective rods (Figure 2(Cc), Figure 3I,J). The transpedicular screw positions were verified with the control of X-ray C-arm (Figure 2(Cd)) and neuromonitoring navigations (Figure 2(Ce,Cf)). Polyaxial and monoaxial transpedicular screws were used; the two corrective rods (5.5 mm in diameter) were made of titanium alloy. The data mining in Stage II of the subject selection for this study (Figure 1) was strictly subordinated to the regime so that in the future analysis, patients of both groups (“Interactive S-N” vs. “Real-time” neuromonitoring) met the same criteria of the surgical treatment technique.

In general, the only difference in the analyzed treatment procedures between both groups of patients was the neuromonitoring scenario (Figure 4). The scoliosis correction was a result of the following maneuvers: rod rotation on the convex side; apical translation; segmental derotation; distraction on the concave side; and compression on the convex side. The wound was closed over in layers. The drain was applied subfascially.

All the TMS-induced MEP recordings (Figure 2A) were taken pre- (a day before the surgery (T0), Figure 3C,D) and postoperatively (a week after the surgery (T3), Figure 3G,H) with the patients in a supine position and in the same diagnostic room with a controlled temperature of 22 °C using the Key-Point Diagnostic System (Medtronic A/S, Skøvlunde, Denmark). A single, biphasic, 5 ms magnetic stimulus generated via the MagPro X100 (Medtronic A/S, Skøvlunde, Denmark) was applied transcranially (Figure 2(Ab)) with a circular coil (C-100, 12 cm in diameter) placed over the scalp in the M1 motor cortex area primarily responsible for the innervation of the lower- and upper-extremity muscles.

The targeted excitation of the corticospinal tract cells of origin and their axons was achieved when the coil was placed exactly perpendicular to the skull surface. It is likely that the cells of origin of the rubrospinal tract in the midbrain can also be excited because the magnetic stream may reach 3 cm deep [32,45]. The MEP measures were used to assess the ability of the primary motor cortex to output neuronal impulses and to evaluate the global efferent transmission of the neural impulses to the effectors via the spinal cord descending tracts. The strength of the magnetic field stream at 70–80% of the resting motor threshold (RMT) (0.84–0.96 T) was applied. The consecutive movements of the magnetic coils distanced 5 mm from each other allowed for assessing the location of the optimal stimulation, a “hot spot” in the area where the TMS elicited the largest recorded MEP amplitude (Figure 2(Ac)). The accurate photographic documentation of the “hot spots” marked at a similar location of the transcranial stimulation allowed for the reproducibility of similar MEP recordings at T0–T3. The TMS was not reported by the subjects as painful. Epileptic side effects were not observed.

Preoperatively (the T0 period of observation), the MEP was recorded with surface electrodes bilaterally from the TA muscles or PER nerves (Figure 2(Aa)) with a pair of disposable Ag/AgCl surface electrodes (5 mm^2^ of active surface). During recordings from the TA muscle, the active electrode was placed on the muscle’s belly, a reference electrode was placed on the muscle’s distal tendon, and a ground electrode was placed in its vicinity. During recordings from the PER nerve, the active electrode was placed proximally, and a reference electrode was placed distally, both at the knee level, lateral to the head of the fibula, in the place of the PER longitudinal, anatomical course. The mountings of the electrodes for the neurophysiological recordings were performed with patients in a supine position during the T0, T1, and T3 observation periods. The resistance between the electrode surfaces and the skin was decreased with electro-conductive gel. The MEP outcome measures were the amplitude in microvolts from peak to peak of the signal and the latency in milliseconds from the stimulus application marked by the artifact in the recording to the onset of the positive inflection of the potential. For the MEP acquisition, the low-pass filter of the recorder was set to 20 Hz, the high-pass filter was set to 10 kHz, the time base was set at 10 ms/D, and the signal amplification was between 200 and 5000 µV. During recordings, a bandwidth from 10 Hz to 1000 Hz, digitalization at 2000 samples per second, and a channel were used.

The results of the two types of intraoperative neuromonitoring (IONM) sessions in the theater using the ISIS system (Inomed Medizintechnik, Emmendinger, Germany) were analyzed in this study. All IONM protocols included detailed data on demographic, anthropometric, and scoliosis characteristics, including the type, range, and angles of the pathological curvature (Table 1), time and type of each activity performed by the anesthesiologists and surgeons, screenshots of MEP recordings with (Figure 3F, in “Real-time” IONM sessions based on simultaneous MEP recordings with the direct visual inspection of the operation field via the camera, without verbal reports) and without (Figure 3E, in “Interactive S-N” IONM sessions based mainly on bilateral surgeon–neurophysiologist verbal reports) the surgical field camera photographs, BIS, stimulus strength for the TES parameters, frequency of neurophysiologist’s warnings and anesthesia-related events, incidence and source of false alarms, averaged time of the surgery, and number of bidirectional communications between the surgeon and neurophysiologists and vice versa. These were used for subsequent analyses of the variables and events associated with intraoperatively recorded MEP parameter fluctuations.

Intraoperatively (the T1 and T2 periods of observation), the MEP was recorded following the application of the transcranial trains of electrical stimuli (TES) in areas of the cortical motor fields in the innervation areas of the selected muscles in the upper and lower extremities. A sequence of four stimuli were applied, with a duration of a single pulse of 500 µs and an intensity of 98.2 mA on average, via a pair of bipolar subcutaneous needle electrodes (Figure 2(Bc)). The impendance of the scalp electrode was about 0.6–0.8 kΩ. The positioning of the stimulating electrodes was based on a compilation of descriptions by Deletis [36] and Legatt et al. [46] according to the 10–20 system; Cz–C3 3–6 cm to the left, and Cz–C4 3–6 cm to the right. The surface disposable Ag/AgCl electrodes (5 mm^2^ of active surface) for the MEP recordings from the TA muscle and PER nerve as well as from the other upper- and lower-extremity muscles (abductor pollicis brevis, extensor carpi group, rectus femoris, abductor hallucis longus), according to the previous descriptions, were successfully used (Figure 2(Ba,Bb)) [8,23]. MEP recordings from the TA muscle as a marker for the purposes of this study were subjected to detailed analysis due to the possibility of comparing their parameters with the descriptions of other researchers monitoring motor conduction intraoperatively in patients with IS. The only other used needle electrode was the ground electrode that was sterilely inserted at the iliac crest (Figure 2(Bb)). The stimulating and recording electrodes were mounted and their connections were checked with the patients in the supine position, and the first MEP recordings were performed as referenced. Later MEPs were recorded with patients in the prone position (the T1 observation period) after the patient was transferred to the operating table (Figure 2(Ca)) at every stage of the surgical scoliosis correction until it was completed (the T2 observation period). Intraoperatively recorded MEPs (Figure 3E,F) were characterized by a variable amplitude from 100 to 2000 µV and latencies in the range of 27–40 ms, but they did not require averaging. The settings of the recorder for the measurements were as follows: high-pass filter hardware: 30 Hz; high-pass software: 0.5 Hz; low-pass software: 2000 Hz; stimulation frequency (Hz): 0.5–2.4 ms intervals.

### 2.3. Statistical Analysis

Data were analyzed and compared using Statistica, version 13.1 (StatSoft, Kraków, Poland). During the preliminary data mining, an attempt was made to match patients from the two groups in this study and the healthy volunteers in terms of their age, sex, and basic anthropomorphic characteristics, such as weight and height, as well as their numbers (Table 1). Minimal and maximal values (range) with means and standard deviations (SDs) were included in the descriptive statistics. The Shapiro–Wilk test and Levene’s test were used to ascertain the normality distribution and the homogeneity of the variances. Bispectral index data and the MEP stimulus strengths were of the ordinal-scale type, while the MEP amplitudes and latencies were of the interval-scale type. None of the collected data represented a normal distribution; the Wilcoxon’s signed-rank test was used to compare the differences between results obtained before (T0) and after (T3) surgeries, as well as to compare results at the beginning (T1) and end (T2) of the surgical procedures. In the cases of independent variables, the non-parametric Mann–Whitney test was used. Any *p*-values < 0.05 were considered statistically significant. The results from all neurophysiological tests performed on patients were also calculated from the group of healthy subjects (control group) to achieve the normative parameters used to compare the health statuses between the patients and controls. The results did not reveal any significant differences in the parameter values recorded in the neurophysiological tests on the left and right sides in the controls. Statistical software, Statistica, version 13.1 (StatSoft, Kraków, Poland) was used to determine the required sample size using the primary outcome variable of the MEP amplitudes recorded from TA muscles before and after treatment with a power of 80% and a significance level of 0.05 (two-tailed). The mean and standard deviation (SD) were calculated using the data from the first 40 patients, and the sample size software estimated that more than 50 patients in each group with different neuromonitoring scenarios were needed for the purposes of this study to observe statistically significant differences.

## 3. Results

The subjects belonging to two groups of patients and the healthy volunteer control group did not differ in demographic and anthropometric characteristics, nor in the severity or extent of the idiopathic scoliosis in cases of patients (Table 1). The number of subjects in each group was similar. These variables and factors probably did not influence the differences found when comparing the neurophysiological results in the studied groups of patients pre-, intra-, and postoperatively, nor those found in the comparison with the controls (Table 2).

In the group of healthy people (control), the amplitude and latency parameters recorded from the TA muscle and PER nerve did not differ significantly when comparing their values on the right and left sides. However, the cumulative parameters of the MEP amplitudes recorded from the PER nerve compared to those recorded from the TA muscle were approximately 1100 µV lower (67%) at *p* = 0.007 (Figure 5A). The cumulative MEP latency parameter recorded bilaterally from the PER nerve compared to the recording from the TA muscle showed 3.1 ms lower values (10.6%) at *p* = 0.04 (Figure 5B).

An overall decrease in the amplitude and a shortening of the latency in the MEP recordings from the PER nerve were also observed in patients from both groups at each follow-up stage at *p* = 0.008–0.04 (Table 2, Figure 4).

In the preoperative examinations (T0), the parameters of the TMS-induced MEP amplitudes recorded from the TA muscle on the right and left sides differed significantly at *p* = 0.04 in both groups of patients but did not when recorded from the PER nerve. Bilateral MEP recordings from the PER nerve at T0 differed similarly in both groups, and only significantly in the latency parameter, which was increased on the left side at *p* = 0.04. In general, patients of both groups with MEP recordings from the TA muscle and PER nerve differed similarly at the T0 and T3 observation stages compared with the healthy volunteers in terms of a decrease in the amplitude parameters and an increase in the latency at *p* = 0.008–0.04. In both groups of patients, the MEP parameters induced by TMS (T0) and TES (T1) did not differ significantly (Table 2, Figure 4). The parameters of the MEP amplitudes recorded both from the TA muscle and PER nerve intraoperatively at T1 and T2 differed significantly at *p* = 0.04–0.03, indicating an improvement in the overall efferent conduction as a result of the surgical intervention for scoliosis correction. One week after surgery (T3), the parameters of the TMS-induced MEP amplitudes further increased bilaterally, compared to the tests recorded at T0 in the range of *p* = 0.03–0.02.

The proper positioning coincidence of the electrodes transcranially stimulating the motor centers for the innervation of first the lower and then the upper muscles using measurements according to the 10–20 system with the method of the preoperative determination of the “hot spots” for the recording of MEPs with the largest amplitudes was calculated at 85%.

The patients of both groups with different spinal efferent transmission neuromonitoring scenarios were optimally anesthetized at the same 51.8 BIS level, on average (Table 3). Similar to the results of our previous studies [8], the present observations indicate that this level of anesthesia, when kept constant, minimally affects the variability in the MEP amplitude parameter when applying a TES stimulus strength at an average of 98.2 mA. We did not observe any significant differences in the variability in the MEP amplitude parameter when we applied both variable values mentioned above in the population of patients from the “Interactive S-N neuromonitoring” and “Real-time neuromonitoring” groups.

The number of MEP parameter fluctuations, primarily decreases in amplitude, was strictly related to the number of events during certain steps of the surgeries and the associated number of warnings from the neurophysiologist (Table 3). They appeared most frequently, almost twice as much, during the transpedicular screw implantation, corrective rod implantation, and distraction, derotation, and compression procedures, and more significantly at *p* = 0.04–0.03 in patients from the “Interactive S-N neuromonitoring” group than in those from the “Real-time neuromonitoring” group. Other changes (5%) in the MEP parameters were detected during surgical field preparation, among others, as the effects of transient “warming” during cauterization (the latency increase) and shocks caused by releasing the vertebral joints (the amplitude decrease). They were observed at almost the same frequency in both groups of treated patients. Overheating of the tissues accompanying the cauterization occasionally caused the temporary slowing down (increase in the MEP latency parameter) of the conduction of nerve impulses in the spinal cord pathways within the white-matter funiculi. The surgical area was rinsed with 0.9% NaCl solution at 36.6 °C in these cases. After the suction of the fluid was applied, this symptom retreated.

Changes in anesthesia levels occurred only occasionally, with no significant differences in the frequencies between the patients of the two studied groups (Table 3). If detected, they had a stronger influence on the amplitudes of the MEPs recorded from the TA muscle than on those recorded from the PER nerve bilaterally. A comparison of recordings presented in Figure 6 may lead to the conclusion that MEPs recorded from nerves undergo significant amplitude fluctuations at different stages of the scoliosis correction only following the surgical interventions, and the anesthesia-level changes have greater meaning for the MEP parameter fluctuations recorded from the muscles than those recorded from the nerves.

The two rarest reasons for warnings during neuromonitoring, with no significant differences, were false alarms caused by technical malfunctions, like electrode resistance changes or disconnections, and movement-related artifacts following TES, which influenced the MEP amplitude changes (Table 3).

The average total duration of the operation, measured from the moment of the initial administration of anesthesia and intubation of the patient to the moment of the transfer from the operating table to the bed after the suturing of the surgical wound (sterile stimulating and grounding electrodes were removed in the meantime) was significantly (*p* = 0.04) shorter by about 1 h in the patients from the “Real-time neuromonitoring” group. The number of two-way communications during surgery in patients from the “Real-time neuromonitoring” group between the surgeon and neurophysiologist and vice versa was significantly (at *p* = 0.008) reduced by approximately half compared to the procedures in patients from the “Interactive S-N neuromonitoring” group (Table 3).

## 4. Discussion

The main methodological finding in this study for the clinical purposes regarding TMS- and TES-induced MEP in healthy volunteers and patients with IS demonstrates the utility of recordings from the nerves. It has been reported that the quality of the MEP recordings from muscles during intraoperative neuromonitoring can be significantly influenced by the depths of the anesthesia or muscle relaxant administration [44]. The possible consequences are the decreased neuronal transmission along the ascending and descending tracts at the spinal and supraspinal levels and the blockade of the transmission of the acetylcholine release at the level of the neuromuscular junction. Moreover, MEP recordings from muscle may fluctuate more than nerve recordings, mainly due to motion artifacts resulting from the aftermath of stimulation. It should also be remembered that the natural, gradual attenuation of the signals may occur more in children than adults during prolonged neurosurgical procedures; the origin of these changes remains unexplained [23]. The results at the different observation stages in this study indicate that although the amplitudes of the recordings from the PER nerve are significantly lower than those recorded from the TA muscle, they are resistant to variability in the level of anesthesia and other factors that inevitably occur during the surgery. Our study provides evidence that, in both groups of patients, an anesthesia level at an average BIS value of 51,8 minimally affects the variability in the MEP amplitude, especially in PER recordings when the applied TES strength is 98.2 mA. MEPs recorded from nerves appear stable enough to provide reliable information regarding the efferent neural conduction in patients with advanced neurogenic muscle changes. The recording of the evoked potential from above the nerve in the place of its anatomical course does not significantly interfere with the recording of the bioelectric activity of the contracting muscle, as shown by the research of Garasz et al. [38]. Recording from the peroneal nerve to verify the overall efferent conduction of nerve impulses from the level of the upper motor neuron following TMS has also been successfully used in experimental studies to assess the effectiveness of the regeneration in nerve fibers [47].

The concept of recording the MEP from the nerve during neuromonitoring is mentioned by Gonzales et al. [48], both following cortical stimulation as well as direct spinal cord stimulation. Among the known descriptions are motor evoked potentials recorded from nerves versus muscles following lumbar stimulation with the magnetic field in healthy subjects and patients with disc–root conflicts, provided by Garasz et al. [38]. In their studies, like in this report, the mean values of the MEP amplitudes recorded from the nerves significantly differed from those recorded in anatomically related muscles in the controls and patients: they were about 30% and 51% smaller, respectively. In both groups of subjects, the latencies of the MEPs recorded from the nerves were shorter (about 3.0 ms) than those recorded from the muscles. The non-invasive method of recording MEPs from nerves can help diagnose patients with visible atrophic changes in the muscles and simultaneous symptoms of only slight pathology in the peripheral transmission of nerve impulses, and similarly in cases of advanced IS patients.

In about 13% of the cases of all the patients in this study belonging to both studied groups who underwent spinal deformity correction, the neuromonitoring team needed to alert the surgeon about the fluctuations in the MEP recording parameters. This corresponds well with the previous data provided by Vitale et al. [21]. According to the data of Hicks et al. [49], the risk of incorrect pedicle screw implantation during IS surgery is lower at 4.2%, and the same may apply to the probable disruption of the spinal cord root structures. The estimates of Kwan et al. [50] are much lower at an approximately 0.95% rate of major complications and a 1.32% rate of minor complications; however, it should be taken into account that they primarily concern the surgical treatment of patients with Lenke type 1 curvature. Similar to the observations of the abovementioned authors, in our study, the number of MEP parameter fluctuations, and mainly decreases in amplitude, was strictly associated with the number of neurophysiologist warnings due to transpedicular screw implantation, corrective rod implantation, and the distraction, derotation, and compression procedures, respectively; at *p* = 0.04–0.03, this number was higher for patients from the “Interactive S-N neuromonitoring” group. A similar order of some sources of threats during IONM was reported by Lyon et al. [51], although, in their opinion, the distraction procedure during IS correction could be the most traumatic. We found that the movement-related artifacts following TES was the rarest reason that influenced the MEP amplitude changes during the IONM, like in the study of Yoshida et al. [52]. The fluctuation in the MEP amplitude parameter [53], less than the latency [54], is typical when describing the most frequent reasons for IONM alerts. The presented results from patients with type 2 IS according to Lenke indicate that the total loss of neuromonitoring signals rarely occurs during the distraction and derotation procedures, which supports the observations of Rizkallah et al. [55] and Nagarajan et al. [56]. Our preliminary comparison of the MEP recordings in the T2 and T3 periods indicates that only patients with fluctuations in the MEP amplitude but not the latency parameters at the level of 45% compared to the controls showed moderate unilateral motor deficits postoperatively, which is similar to the finding of Buckwalter et al. [57].

In all patients, the motor-evoked-potential parameters induced by TMS (T0) and TES (T1) did not differ in this study. This indicates the importance of a preoperative neurophysiological assessment, showing the current neurological status of a patient with IS, which should also be expected in the perioperative period for the purposes of neuromonitoring, constituting a source of valuable knowledge for both the neurophysiologist and the surgeon. The same opinion is shared by Glasby et al. [58], Virk et al. [59], and Lo et al. [57], who also underline that the early recognition of the MEP properties is important to prevent false positives during the course of IONM for spinal surgery.

This and previous studies [8,23,24,25] indicate the high sensitivity of the recording surface electrodes when performing IONM, sufficient to ensure the reliable verification of spinal efferent conduction during scoliosis surgery. According to our experience using surface electrodes for IONM purposes, the cost of a set of eight pairs compared to that of needle electrodes is nine times lower [23]. The postoperative disposal of surface electrodes is also safer, more accessible, ecological, and less expensive than that of needle electrodes, and their non-invasiveness is non-disputable.

The pediatric nature of the corrective spine procedures in patients with IS dictates the need for the least amount of invasiveness, which was met in the current work by recording the MEPs using surface electrodes from the muscles and nerves of the lower extremities [60]. For the IONM, we used a pair of needle electrodes (not the standard corkscrew electrodes) applied bilaterally and subcutaneously on the scalp for stimulation purposes, and a single needle electrode at the level of the iliac crest as the ground electrode. Thanks to this choice, we avoided ecchymoses, bruises, and rare infections, which have been reported during IONM [61], as well as reddening of the skin, which was observed in 16% of patients with the accompanying symptom of increased pain, sometimes lasting up to 6 months after the surgery [62], and a significant risk of needle stick injuries by neurophysiologists and other workers in the theater during electrode implantation and their removal after surgery.

In this report, as in the previous work [8], we demonstrated an immediate improvement in the parameters of the nerve impulse conduction of efferent transmission in the spinal pathways following the surgical correction of scoliosis at T2. Similarly, in both groups with different neuromonitoring scenarios, we also found further improvement at T3, based on the bilateral recordings of the MEP amplitudes from the tibialis anterior (TA) muscles, as well as recordings from the PER nerves. Moreover, we found that the average duration of the surgery was significantly shorter by about one hour in the “Real-time neuromonitoring” group, and the number of two-way communications between the surgeon and neurophysiologist and vice versa was reduced by approximately half. This proves that the “Real-time neuromonitoring” scenario shortens the two-way IONM communication between the neurophysiologist and the surgeon performing subsequent scoliosis correction procedures based on the principle of “fewer words—better concentration and faster reaction”, increasing the effectiveness of the neuromonitoring and the working comfort of the surgeon who is solely engaged in performing the IS correction procedures. This is yet more proof that the success of the complex surgical correction of IS with the effective involvement of IONM depends on the close cooperation and optimal coordination of the team of surgeons, anesthesiologists, neurophysiologists, and instrumental staff in the operating room [34].

The guidance states that, from the very beginning of the IONM application, all neurophysiological procedures should be performed and interpreted by a clinical neurophysiologist with the appropriate training and credentials [63,64]. Moreover, “Real-time neuromonitoring” requires the neurophysiologist to have extensive knowledge of the surgical procedures during scoliosis correction, knowledge of the skills and individual behaviors of the surgical team members, and the ability to interpret unexpected events during surgery using good-quality recording equipment and high-resolution images of the surgical field from the camera.

Considering the clinical impact and summarizing the results of this study, the superiority of the “Real-time” IONM concept in the context of increasing safety supports the evaluation of MEP recordings from the nerves, which are more resistant to possible changes in the level and duration of anesthesia and stimulation condition fluctuations than recordings from muscles. The non-invasiveness is supported by the use of surface electrodes for pediatric subjects, and by the shorter duration of the surgery when this type of IONM is performed.

One of the limitations of this study could be that the patients selected for the samples in both groups with different scenarios of IONM represented different types of IS curvatures from 1 to 3 (mainly 2) according to Lenke, which could have biased the study design, especially the recorded parameters of the MEP in the T0–T3 periods of observation. In other words, the potential influence of the different types and severities of scoliosis due to selection bias may have influenced the TES-induced MEP results. We attempted to eliminate this limitation by selecting subjects with similar types and severities of scoliosis and, in cases of patients belonging to the two studied groups, similar extents of idiopathic scoliosis (see Figure 1, Stage I). The main type of scoliosis with similar numbers in both groups was Lenke 2 (N = 48 vs. N = 46), with a few cases of Lenke 1 and Lenke 3 occurring in the same proportions; no statistical differences were detected in the primary and secondary angles of the spine curvatures (see Table 1, bottom). It is unlikely that these variables in both groups of patients could have influenced the final MEP recording results and, therefore, the differences found in the two groups with different neuromonitoring scenarios.

The disadvantages of using surface electrodes for IONM purposes may include the technical aspects of their greater resistance than that of needle electrodes and their possible dislocation from the source of the bioelectric signal, successfully resolved with the addition of sterile mounting tape (Figure 2(Bb,Bc)). Moreover, the recorded bioelectric signals are characterized by lower amplitudes. However, this study provides evidence that these factors do not significantly influence the quality of MEP recordings.

Correlations between the recorded MEP parameters and results of the detailed clinical studies, including neurological evaluations, are not presented; we never intended to include clinical evaluations of the IS patients in our project, nor is this reported as the aim of the presented study. No detailed evaluation using the classical neurological methods of IS patients has been provided in previous studies so far. This is a good reason for future research on the neurological outcomes of the surgical treatment of IS.

Future neurophysiological studies should also concentrate on the correlation of the morphologies and parameters of MEPs recorded pre- and postoperatively with the clinical evaluation of the curvature angle improvement, which has not been studied in detail in IS patients and has not only cognitive but also practical meaning.

## 5. Conclusions

Although the presented concept of “Real-time neuromonitoring” seems obvious, the results of its application may lead to the conclusion that the cooperation of the neurophysiologist with the surgeon with minimal two-way verbal communication during IONM increases the surgeon’s attention to surgical procedures only, and increasing the surgeon’s working comfort affects the safety of the entire surgery. “Real-time neuromonitoring” shortens the overall duration of the surgery, which minimizes the side-effect impact of the anesthesia administered to the patient, which is of the greatest importance, considering its long-term effects on the cardiac and nervous system functions. The modifications of the MEP nerve-conduction-recording technology presented in this study using surface electrodes on the nerves provide precise and reliable information about the patient’s neurological condition at every stage of the surgical procedure, even under conditions of slight fluctuations in anesthesia. It is non-invasive and more economical than standard needle recordings and can be used for IONM during surgery on pediatric patients, including patients with IS.

## Figures and Tables

**Figure 1 jcm-13-01497-f001:**
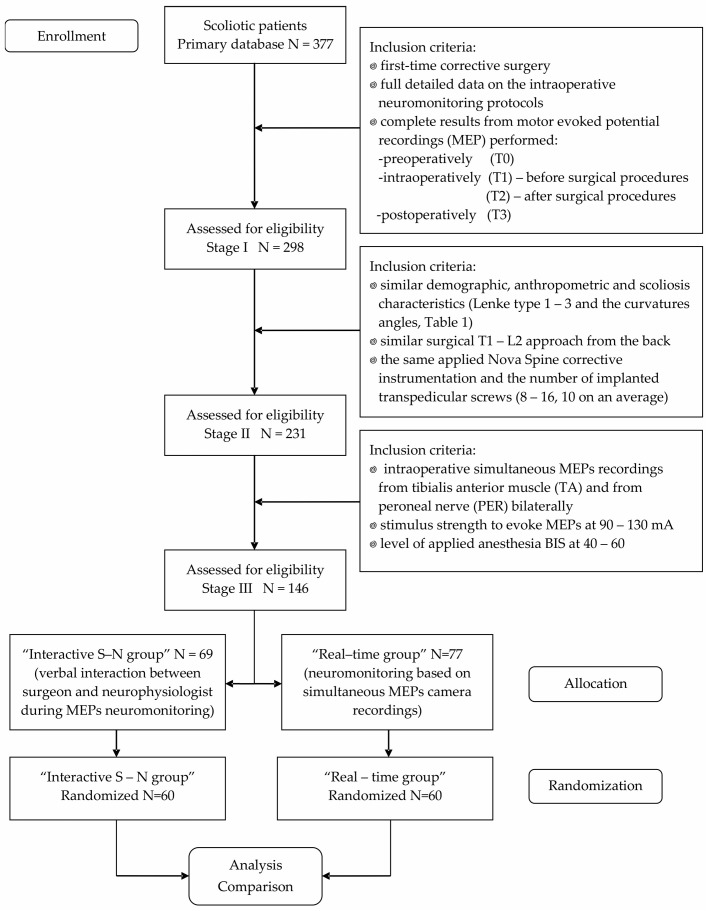
Flowchart of study with design and scoliotic subject selection criteria at subsequent three stages. Abbreviations: MEPs—motor evoked potentials.

**Figure 2 jcm-13-01497-f002:**
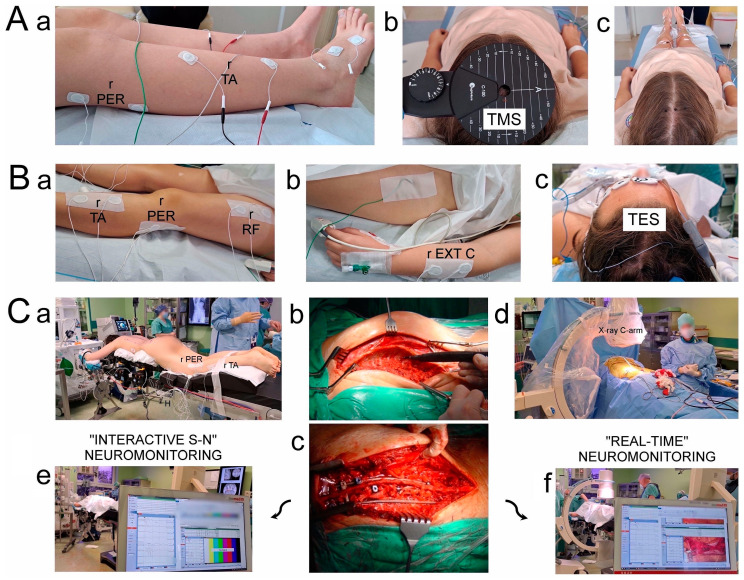
Methodological principles of neurophysiological recordings and study design. The same neurophysiological methodology was used in healthy volunteers once. (**Aa**)—Location of recording bipolar electrodes over anatomical passage of peroneal (PER) nerve and surface of tibialis anterior (TA) muscle applied in pre- and postoperative examinations. (**Ab**)—Preoperative positioning of stimulating coil over scalp for transcranial magnetic stimulation (TMS) (experimentally changed following tracking with aim of obtaining best highest-amplitude MEP recordings), which allowed marking black points of “hot spots” (**Ac**) for intraoperative stimulating electrode application ((**Bc**) for transcranial electrical stimulation (TES)). Placement of intraoperative recording bipolar electrodes over TA muscle and PER nerve, as well as rectus femoris (RF) muscle (**Ba**), and over forearm extensor carpi (EXT C) muscle group (**Bb**), was stabilized with adhesive patch tapes. (**Ca**)—Prone position of patient with applied recording electrodes prepared for surgery from back approach in theater. Photographs illustrating view of thoracolumbar spine preparation at subsequent steps before (**Cb**) and after (**Cc**) implantation of two titanium rods for distraction and derotation procedures performed by surgeon. Positioning of transpedicular screws was also verified via X-rays with C-arm (**Cd**). Neuromonitoring recordings in distant range from surgical field aimed at verifying spinal neural motor transmission without ((**Ce**), in “Interactive S-N” patient group) or with ((**Cf**), in “Real-time” patient group) camera picture support and simultaneous MEP recordings. Abbreviations: r—recording; TA—tibialis anterior muscle recording; PER—peroneal nerve recording; TMS—transcranial magnetic stimulation; TES—transcranial electrical stimulation.

**Figure 3 jcm-13-01497-f003:**
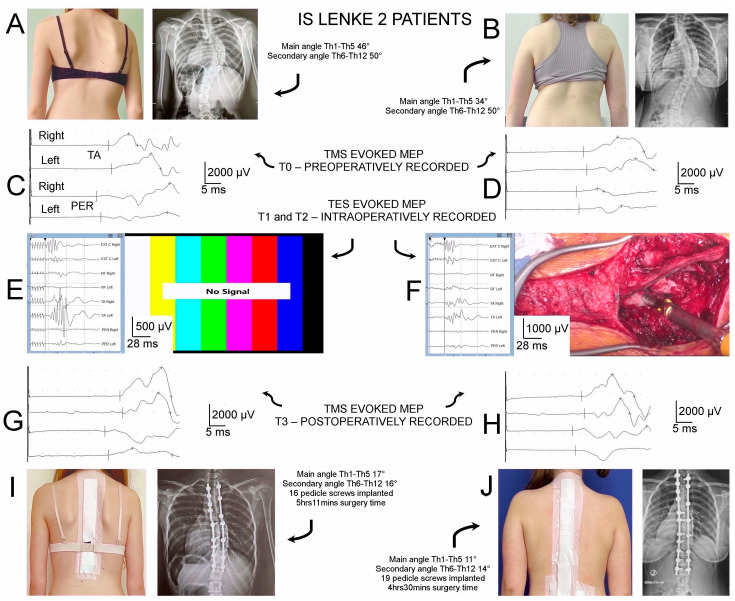
Examples of most relevant recordings from X-ray clinical and MEP neurophysiological studies in patients of two studied groups. Photographs of body silhouettes and anterior–posterior X-rays performed at T0 in patients belonging to both groups are shown in (**A**,**B**), as well as in (**I**,**J**) at T3, for comparison of similarities, respectively. Note the diminishing of the primary and secondary spine curvatures. Examples of MEPs recorded at T0 (**C**,**D**) and T3 (**G**,**H**) following TMS and intraoperatively following TES ((**E**) vs. (**F**)) presented for comparison to indicate increasing amplitudes in patients of two studied groups (both with Lenke type 2 scoliosis), especially in PER recordings. Calibration bars for amplification (vertical) and time base (horizontal) of MEP recordings set during neurophysiological tests are shown. Note that amplifications of recordings in (**E**) are greater than those in (**F**). Abbreviations: IS—idiopathic scoliosis; T0—preoperative period of observation; T1—intraoperative period of observation before surgical procedures; T2—intraoperative period of observation after surgical procedures; T3—postoperative period of observation after one week; TA—tibialis anterior muscle recording; PER—peroneal nerve recording; MEP—motor evoked potential; TMS—transcranial magnetic stimulation; TES—transcranial electrical stimulation.

**Figure 4 jcm-13-01497-f004:**
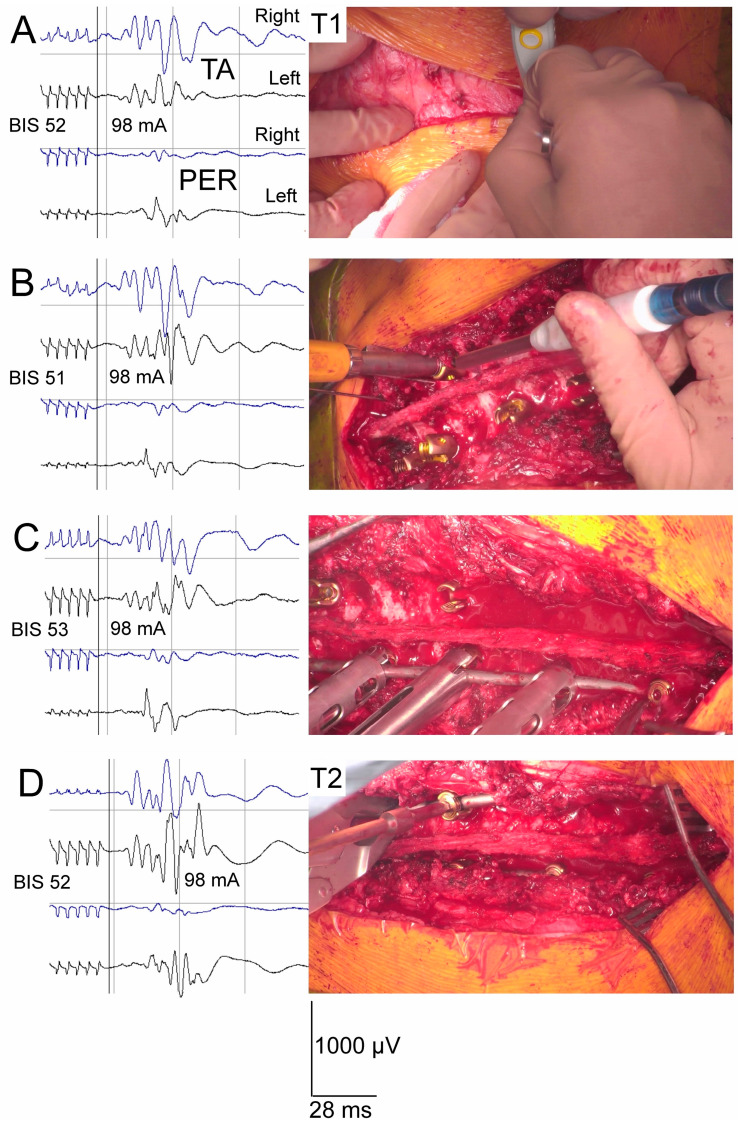
Methodological concept of “Real-time intraoperative neuromonitoring”, when changes in parameters of MEP bilateral recordings are analyzed during, but not after, performing subsequent steps of surgical procedures that could insult spinal cord structures. Observation of surgical field with camera picture allows for an immediate warning reaction when MEP parameters change simultaneously. (**A**)—Surgical field preparation with muscle cauterization; (**B**)—transpedicular screw implantation; (**C**)—corrective rod implantation and correction, distraction, and derotation of spine curvature; (**D**)—contralateral corrective rod fixation. Calibration bars for amplification (in µV) and time base (in ms) are the same for each MEP recording. Note the more stable nerve recordings at every step of the surgery, even at slight changes in anesthesia level verified by BIS. Abbreviations: BIS—bispectral index monitor; TA—tibialis anterior muscle recording; PER—peroneal nerve recording; T1—period of observation before performance of surgical procedures; T2—period of observation after completion of surgical procedures.

**Figure 5 jcm-13-01497-f005:**
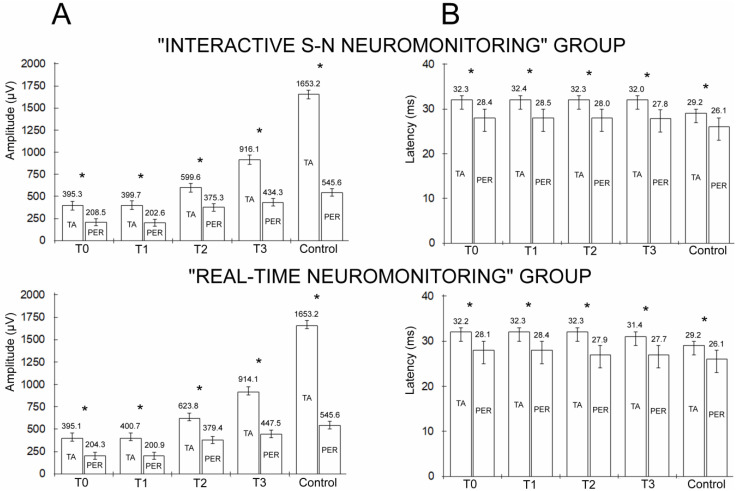
Comparison of amplitude (**A**) and latency (**B**) cumulative values of MEPs recorded from tibialis anterior (TA) muscles and peroneal (PER) nerves in patients belonging to two groups with different neuromonitoring scenarios in four periods of observation (T0—preoperative; T1—intraoperative before IS correction; T2—intraoperative after IS correction; T3—postoperative). Normative parameters are presented for comparison as well (control). Abbreviation: * *p* < 0.05 determines significant statistical differences.

**Figure 6 jcm-13-01497-f006:**
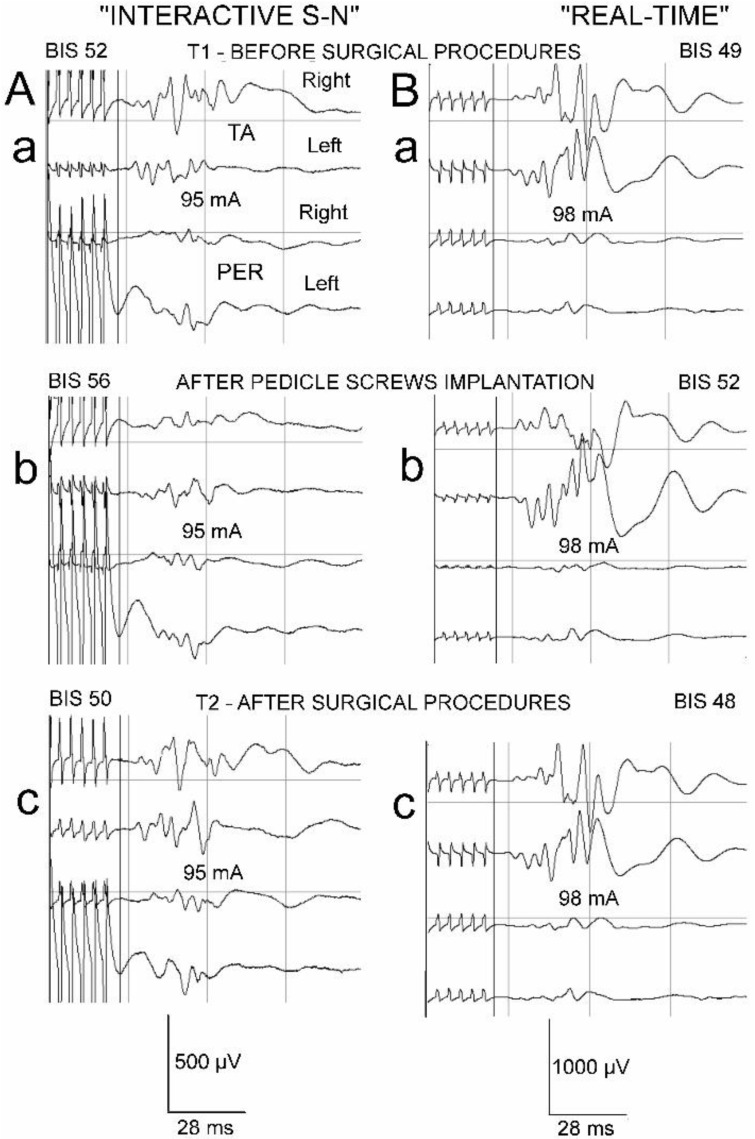
Examples of intraoperative MEP recordings in two patients from “Interactive S-N” (**A**) and “Real-time” (**B**) groups undergoing surgical correction of idiopathic scoliosis before surgical procedures (T1, (**a**)), after pedicle screw implantation (**b**), and after completion of all surgical procedures (T2, (**c**)). The arrangement of recording sites from muscles and nerves bilaterally presented in Aa is common for all examples. Note that calibration bars for amplifications (in µV) are different for recordings in (**A**,**B**). Recordings were performed in two patients at similar levels of anesthesia as indicated by BIS values and following similar strengths of transcranial stimulation expressed in mA. Comparison of amplitudes in MEP recordings, especially from nerves, evidences the resistance to anesthetic condition changes: they only fluctuate following surgical procedures. Abbreviation: BIS—bispectral index monitor; TA—tibialis anterior muscle recording; PER—peroneal nerve recording; T1—period of observation before performance of surgical procedures; T2—period of observation after completion of surgical procedures.

**Table 1 jcm-13-01497-t001:** Demographic, anthropometric, and pathological scoliosis characteristics in patients from two study groups and healthy volunteer controls. Ranges, mean values, and standard deviations are presented. *p* < 0.05 determines significant statistical differences.

Variable Group of Subjects	Age (Years)	Height (cm)	Weight (kg)	BMI	Scoliosis Type [42]	Cobb’s Angle [43] (Preoperatively)
“Interactive S-N” neuromonitoring group N = 60 ♀	8–17 14.2 ± 1.6	135–181 164.4 ± 2.0	30–84 54.5 ± 2.9	17.6–29.9 23.1 ± 4.0	Lenke 1 = 10 Lenke 2 = 48 Lenke 3 = 2	Primary 41–86 57.4 ± 6.6 Secondary 24–50 37.1 ± 3.3
“Real-time” neuromonitoring group N = 60 ♀	9–18 14.7 ± 1.5	137–179 165.6 ± 2.5	29–83 53.1 ± 3.1	17.4–30.1 22.9 ± 3.9	Lenke 1 = 11 Lenke 2 = 46 Lenke 3 = 3	Primary 40–89 56.3 ± 7.1 Secondary 25–50 37.3 ± 3.9
Healthy volunteer “Control” group N = 60 ♀	8–18 14.3 ± 1.5	134–183 166.1 ± 2.6	30–84 54.9 ± 5.3	17.4–29.8 22.8 ± 3.7	NA	NA
*p*-value (difference) “Interactive S-N” vs. “Real-time” “Interactive S-N” vs. “Control” “Real-time” vs. “Control”	0.223 NS 0.177 NS 0.082 NS	0.182 NS 0.192 NS 0.091 NS	0.171 NS 0.122 NS 0.079 NS	0.183 NS 0.089 NS 0.119 NS	0.062 NS	Primary angle 0.199 NS Secondary angle 0.328 NS

Abbreviations: ♀—female; “Interactive S-N group”—verbal interaction between surgeon and neurophysiologist during intraoperative neuromonitoring continuously maintained; “Real-time group”—intraoperative neuromonitoring mainly based on simultaneous recording and inspection of evoked potential and camera recordings by neurophysiologist; NS—non-significant; NA—not applicable.

**Table 2 jcm-13-01497-t002:** Comparison of results from motor-evoked-potential recordings performed in two groups of 120 IS patients pre- (T0), intra- (before (T1) and after (T2) surgical procedures), and postoperatively (T3) and 80 healthy volunteers (controls). Ranges, means, and standard deviations are presented. *p* < 0.05 determined significant statistical differences, marked in bold.

	Test Parameter	Side	TMS Control N = 60	Scoliosis Side	TMS Patients Preoperative T0	Control vs. Patients T0	TES Patients Intraoperative T1 (Before IS Correction)	TMS Patients T0 vs. TES Patients T1	TES Patients Intraoperative T2 (After IS Correction)	TES Patients T1 vs. T2	TMS Patients Postoperative T3	TMS Patients T0 vs. T3	Control vs. Patients T3
			Min–Max. Mean ± SD		Min.–Max. Mean ± SD	*p*-Value	Min.–Max. Mean ± SD	*p*-Value	Min.–Max. Mean ± SD	*p*-Value	Min.–Max. Mean ± SD	*p*-Value	*p*-Value
**MEP recorded from tibialis anterior (TA) muscle**													
**“INTERACTIVE S-N”** **NEUROMONITORING** **GROUP N = 60**	Amplitude (µV)	R	1300–3600 1695.1 ± 92.8	Convex	250–1400 412.1 ± 70.4	**0.009**	200–1300 430.4 ± 78.1	0.091	400–1850 688.2 ± 76.4 ↑	**0.029**	700–2400 977.1 ± 99.1 ↑	**0.023**	**0.008**
		L	1000–3050 1611.9 ± 72.8	Concave	200–1300 385.4 ± 49.8	**0.009**	150–1050 369.9 ± 73.6	0.094	300–1650 511.1 ± 78.3 ↑	**0.042**	550–1900 855.3 ± 100.2 ↑	**0.019**	**0.008**
	*p*-value	R vs. L	0.119	Convex vs. Concave	**0.048**	NA	**0.047**	NA	**0.045**	NA	**0.048**	NA	NA
	Latency (ms)	R	24.3–31.6 28.8 ± 1.4	Convex	27.2–36.1 31.9 ± 3.1	**0.036**	28.9–38.1 31.7 ± 1.5	0.121	28.0–38.3 31.4 ± 1.4	0.299	28.5–39.1 30. 9 ± 2.0	0.058	**0.031**
		L	25.1–32.0 29.6 ± 1.5	Concave	28.8–39.1 32.7 ± 2.6	**0.037**	29.4–39.6 32.9 ± 2.0	0.111	30.3–40.2 33.2 ± 2.6	0.298	30.5–40.0 33.2 ± 2.1	0.060	**0.041**
	*p*-value	R vs. L	0.205	Convex vs. Concave	0.077	NA	0.052	NA	0.067	NA	0.058	NA	NA
	**MEP recorded from peroneal (PER) nerve**												
	Amplitude (µV)	R	450–2050 565.7 ± 55.4	Convex	100–800 213.2 ± 46.1	**0.028**	100–700 218.1 ± 44.3	0.114	200–800 ↑ 388.3 ± 39.8 ↑	**0.043**	200–800 ↑ 443.6 ± 33.8 ↑	**0.034**	**0.044**
		L	400–2000 525.7 ± 58.2	Concave	50–700 186.2 ± 36.3	**0.029**	50–600 187.2 ± 40.1	0.009	300–750 362.3 ± 38.1	**0.041**	350–800 425.1 ± 41.4	**0.031**	**0.045**
	*p*-value	R vs. L	0.112	Convex vs. Concave	0.102	NA	0.052	NA	0.073	NA	0.072	NA	NA
	Latency (ms)	R	22.1–28.9 25.9 ± 1.6	Convex	23.0–31.4 27.6 ± 3.4	**0.042**	22.9–31.1 27.8 ± 3.5	0.075	22.7–31.0 27.8 ± 3.3	0.321	22.5–31.3 27.6 ± 3.2	0.091	**0.041**
		L	22.9–30.0 26.3 ± 1.6	Concave	23.6–33.5 29.2 ± 3.6	**0.039**	23.7–34.0 29.1 ± 3.3	0.111	23.4–32.1 28.4 ± 3.6	0.114	22.9–32.0 28.0 ± 3.1	0.061	**0.046**
	*p*-value	R vs. L	0.095	Convex vs. Concave	**0.048**	NA	**0.049**	NA	0.051	NA	0.073	NA	NA
TA vs. PER cumulative (R+L) MEP amplitude difference (µV)			1108.2 ↓		189.8 ↓		197.5 ↓		224.3 ↓		481.8 ↓		
% of difference			67.1 ↓		47.8 ↓		49.3 ↓		37.4 ↓		52.5 ↓		
*p*-value			**0.007**		**0.009**		**0.009**		**0.022**		**0.008**		
TA vs. PER cumulative (R+L) MEP latency difference (µV)			3.1 ↓		3.9 ↓		3.9 ↓		4.3 ↓		4.2 ↓		
% of difference			10.6 ↓		12.0 ↓		12.0 ↓		13.3 ↓		13.1 ↓		
*p*-value			**0.043**		**0.032**		**0.032**		**0.031**		**0.030**		
	**MEP recorded from tibialis anterior (TA) muscle**												
**“REAL-TIME” NEUROMONITORING GROUP N = 60**	Amplitude (µV)	R	1300–3600 1695.1 ± 92.8	Convex	300–1350 410.9 ± 62.1	**0.008**	250–1400 425.7 ± 71.2	0.084	450–1900 696.9 ± 69.3 ↑	**0.027**	750–2300 975.4 ± 65.3 ↑	**0.021**	**0.018**
		L	1000–3050 1611.9 ± 72.8	Concave	150–1250 380.1 ± 45.4	**0.009**	200–1200 375.8 ± 73.6	0.085	350–1950 550.7 ± 59.8 ↑	**0.038**	600–1850 852.9 ± 88.1 ↑	**0.018**	**0.019**
	*p*-value	R vs. L	0.119	Convex vs. Concave	**0.047**	NA	**0.046**	NA	**0.040**	NA	**0.047**	NA	NA
	Latency (ms)	R	24.3–31.6 28.8 ± 1.4	Convex	27.7–37.3 31.6 ± 3.5	**0.032**	28.8–38.6 31.8 ± 2.8	0.185	28.1–38.6 31.3 ± 2.5	0.119	28.2–39.0 31. 1 ± 2.5	0.059	**0.034**
		L	25.1–32.0 29.6 ± 1.5	Concave	28.7–38.9 32.8 ± 3.1	**0.034**	29.6–39.2 32.8 ± 3.2	0.206	30.4–41.9 33.4 ± 2.9	0.194	30.1–39.8 31.8 ± 2.8	0.061	**0.038**
	*p*-value	R vs. L	0.205	Convex vs. Concave	0.081	NA	0.055	NA	0.061	NA	0.251	NA	NA
	**MEP recorded from peroneal (PER) nerve**												
	Amplitude (µV)	R	450–2050 565.7 ± 55.4	Convex	150–900 218.4 ± 45.8	**0.031**	150–750 215.9 ± 48.2	0.123	300–900 390.8 ± 35.2 ↑	**0.040**	250–900 448.9 ± 36.1 ↑	**0.029**	**0.046**
		L	400–2000 525.7 ± 58.2	Concave	100–800 189.8 ± 48.1	**0.030**	100–650 185.9 ± 38.3	0.008	300–800 368.1 ± 38.9 ↑	**0.043**	300–805 446.1 ± 38.1 ↑	**0.027**	**0.048**
	*p*-value	R vs. L	0.212	Convex vs. Concave	0.108	NA	0.055	NA	0.065	NA	0.080	NA	NA
	Latency (ms)	R	22.1–28.9 25.9 ± 1.6	Convex	23.4–31.8 27.2 ± 2.8	**0.043**	23.2–31.4 27.8 ± 3.5	0.074	22.6–31.4 27.8 ± 3.3	0.325	22.8–31.1 27.5 ± 2.8	0.082	**0.040**
		L	22.9–30.0 26.3 ± 1.6	Concave	23.8–33.3 29.1 ± 3.1	**0.041**	23.5–33.9 29.0 ± 3.1	0.129	23.5–32.0 28.1 ± 3.1	0.121	22.7–32.1 28.0 ± 3.2	0.073	**0.044**
	*p*-value	R vs. L	0.224	Convex vs. Concave	**0.047**	NA	**0.047**	NA	0.055	NA	0.081	NA	NA
TA vs. PER cumulative (R+L) MEP amplitude difference (µV)			1108.2 ↓		191.2 ↓		199.8 ↓		244.4 ↓		466.6 ↓		
% of difference			67.1 ↓		48.3 ↓		49.8 ↓		39.1 ↓		51.0 ↓		
*p*-value			**0.007**		**0.009**		**0.009**		**0.023**		**0.008**		
TA vs. PER cumulative (R+L) MEP latency difference (µV)			3.1 ↓		4.1 ↓		3.9 ↓		4.4 ↓		3.7 ↓		
% of difference			10.6 ↓		12.7 ↓		12.0 ↓		13.6 ↓		11.7 ↓		
*p*-value			**0.043**		**0.033**		**0.032**		**0.028**		**0.048**		

Abbreviations: TMS—transcranial magnetic stimulation; TES—transcranial electrical stimulation; MEP—motor evoked potential; TA—tibialis anterior muscle; PER—peroneal nerve; NA—not applicable; arrows: ↓ indicates a decrease in the parameter when the cumulative values are compared, and ↑ indicates an increase when the MEP amplitude was recorded at T2 and T3.

**Table 3 jcm-13-01497-t003:** Data on variables and events associated with intraoperatively recorded MEP parameter fluctuations in 120 scoliotic patients belonging to two studied groups for comparison. Ranges, means, and standard deviations are presented as well as numbers of incidences. *p* < 0.05 determines significant statistical differences, marked in bold.

Variable Neurophysiological Event Surgical Event	“Interactive S-N Neuromonitoring” Group N = 60	“Real-Time Neuromonitoring” Group N = 60	Difference *p*-Value
BIS level	40–65 (52.4 ± 4.1)	40–60 (51.3 ± 3.9)	0.236
TES strength (mA) during maximal-amplitude MEP recordings	80–124 (99.2 ± 8.1)	80–130 (97.3 ± 7.3)	0.194
Number of neurophysiologist warnings			
associated with MEP parameter fluctuations:			
-During surgical field preparation			
(including effects of transient “warming” during cauterization and shocks caused by releasing vertebral joints)	9/60	7/60	0.07
-During pedicle screw implantation	12/60	8/60	**0.04**
-During corrective rod implantation	10/60	6/60	**0.04**
-During distraction, derotation, compression	8/60	4/60	**0.03**
Anesthesia-related events:			
-MEPS from TA recorded bilaterally	5/60	4/60	0.135
-MEPS from PER recorded bilaterally	1/60	2/60	0.163
False alarms caused by technical malfunctions	1/60	2/60	0.165
False alarms caused by movement-related artifacts following TES	1/60	0/60	0.239
Average time of surgery	5.5	4.5	**0.04**
Number of bidirectional communications:			
-S vs. N	587	292	**0.008**
-N vs. S	396	191	**0.008**

Abbreviations: TES—transcranial electrical stimulation; “Interactive S-N group”—verbal interaction between surgeon and neurophysiologist during intraoperative neuromonitoring continuously maintained; “Real-time group”—intraoperative neuromonitoring mainly based on simultaneous recording and inspection of evoked potential and camera recordings by neurophysiologist; TA—tibialis anterior muscle; PER—peroneal nerve; S—surgeon; N—neurophysiologist.

## Data Availability

All the data generated or analyzed during this study are included in the published article.

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
