# Peer review of "“Real-Time Neuromonitoring” Increases the Safety and Non-Invasiveness and Shortens the Duration of Idiopathic Scoliosis Surgery"

_jcm, 2024, doi:10.3390/jcm13051497_

Round 1

Reviewer 1 Report

Comments and Suggestions for Authors

Please see the attached word file.

Comments on the Quality of English Language

Please look for English corrections while re-submitting the manuscript.

Reviewer 2 Report

Comments and Suggestions for Authors

Authors present a retrospective study on 120 female patients who underwent surgery for adult scoliosis Lemke type 2 to compare two types of neuromonitoring:  "Real-time neuromonitoring (simultaneous neurophysiological recording and the inspection of the surgical field through the camera) " and "Interactive verbal surgeon neurophysiologist neuromonitoring", with 60 healthy controls. The number of two-way communications between the surgeon and neurophysiologist and vice versa in the "Real-time neuromonitoring" group was reduced by approximately half at p=0.008,  increasing safety and non-invasiveness, shortening the time, and lowering the costs of surgical treatment of patients with pathological lateral curvature of the spine.

The manuscript, especiall abstract, is confusely written and needs English language editing. Low number of patients and retrospective character are drawbacks. I suggest to correct Figure 2 and split it into two for better understanding of the two concepts. The real difference between the two methods should be demonstrated on illustrative cases. 

Comments on the Quality of English Language

Extensive English language editing required. 

Reviewer 3 Report

Comments and Suggestions for Authors

This manuscript compared two intraoperative neuromonitoring scenarios, namely "Real-time neuromonitoring" and "Interactive verbal surgeon-neurophysiologist neuromonitoring," in patients undergoing surgical treatment for mainly Lenke 2 type idiopathic scoliosis. By utilizing non-invasive recordings from peroneal nerves (PER) and tibialis anterior muscles (TA) with surface electrodes of motor evoked potentials (MEP) bilaterally, induced by transcranial magnetic (TMS) or electrical (TES) stimulations, the study aimed to assess advantages, utility, reliability, and time consumption of both approaches. The findings reveal that "Real-time neuromonitoring" offers significant advantages, including increased safety, shorter surgical duration, and reduced communication requirements between the surgeon and neurophysiologist compared to the "Interactive verbal surgeon-neurophysiologist neuromonitoring" approach.

The modifications in MEP nerve conduction recording technology presented in the study offer promising advancements in intraoperative monitoring techniques for patients with pathological lateral curvature of the spine.

Overall, the study had a solid design and was conducted carefully. It was well-written overall. This study should be of interest to readers of JCM and researchers in related fields.

 Concerns:

About the abstract:

The abstract provides a comprehensive overview of the research study but could be enhanced by improving the clarity, structure, and presentation of key findings and implications.

1. Consider breaking it down into distinct sections, such as Background, Methods, Results, and Conclusions, to enhance readability and understanding.

2. About the results described in the abstract, it could be more concise and focused on the results or findings. Consider highlighting the key results and their implications.

 About Table 2:

This table provides a comprehensive comparison of motor evoked potential recording performed in two groups of IS patients, as well as the healthy volunteers. It would be helpful to provide brief interpretations or explanations of the observed trends and significant differences to aid readers' understanding.

The inclusion of p-values indicating significant differences (p < 0.05) in bold helps highlight important findings. However, it would be beneficial to include the statistical methods.

 About Figure 4:

Ensure accurate labeling of statistical differences on the bar graph by adding a line and (*) above the two groups when a statistical difference is present. Should include the statistical method in the figure legend.

Reviewer 4 Report

Comments and Suggestions for Authors

While the discussed article presents valuable insights into the utility of nerve recordings in intraoperative neuromonitoring (IONM) during scoliosis surgery, it's essential to consider potential limitations or drawbacks:

1.     The study acknowledges that patients selected for the sample might have represented different types of scoliosis curvatures according to Lenke, which could introduce bias. This limitation may impact the generalizability of the findings.

2.     The article does not provide a detailed analysis of clinical outcomes, such as postoperative neurological deficits or complications. Therefore, the direct correlation between the recorded MEP parameters and clinical outcomes is not thoroughly explored.

3.     The preliminary selection aimed to choose patients with similar primary and secondary spinal curvature angles in both groups, but selection bias could still be present. The potential impact of varying scoliosis types and severity on the results is not fully addressed.

4.     Some aspects of the methodology, such as the detailed principles of nerve recording following direct spinal cord stimulation or the TMS and TES procedures, are not fully described. This lack of detailed information may limit the reproducibility of the study.

5.     While the article mentions comparisons with previous studies, it does not extensively discuss or contextualize the results within the existing literature. A more comprehensive comparison with other relevant studies would provide a better understanding of the significance of the findings.

6.     The article highlights the advantages of "Real-time neuromonitoring," such as shorter surgery duration and reduced communication. However, potential disadvantages or challenges associated with minimizing communication between the neurophysiologist and surgeon are not extensively discussed.

7.     The article mentions that fluctuations in MEP recordings are associated with specific stages of surgery, but it may not account for all potential confounding factors influencing these fluctuations during scoliosis correction procedures.

Round 2

Reviewer 2 Report

Comments and Suggestions for Authors

Authors have sufficiently responded to reviewer remarks. 

Reviewer 4 Report

Comments and Suggestions for Authors

I thank the authors for their corrections and replies. I think it can be published in its current form.